# OpenReview forum: "SecCodePLT: A Unified Platform for Evaluating the Security of Code GenAI"
_ICLR.cc/2025/Conference — Submitted to ICLR 2025_

### Official Review · Reviewer_N2DR · 2024-11-02

**Soundness:** 2
**Presentation:** 2
**Contribution:** 3
**Rating:** 6
**Confidence:** 3

**Summary:**

This paper provides a benchmark for evaluating security issues associated with LLM generated code. Specifically covering:
i) Secure code generation: to assess LLMs ability to generate secure code (focusing on Python).
ii) Cyber attack helpfulness: to evaluate a model’s capability in facilitating end-to-end cyberattacks.
They apply 4 LLMs to both benchmarks -- CodeLlama-34B-Instruct, Llama-3.1-70B, Mixtral-8×22B, GPT-4o – and compare their performance.

**Secure code generation benchmark:**
The authors manually created 153 seed tasks covering 27 CWEs relevant to python – then used LLM-based mutators to generate variations of the tasks for each of the seeds (for large scale generation). They also include both vulnerable and patched code versions, together with functionality and security test cases for each task – resulting in a total of  1345 samples with about 5 test cases per sample.
* They evaluate their samples on ‘prompt faithfulness’ and ‘security relevance’ – comparing with CyberSecEval and outperforming it on both.
* They also evaluate the 4 LLMs for achieving the task’s required functionality using the pass @1 metric on the provided unit tests. And they evaluate the code security using carefully constructed security tests, including the boost in security when providing security policy info in the prompt.
* They also evaluate Cursor on their benchmark.

**Cyber attack benchmark:**
For this, they build a simulated environment containing a network that runs an e-commerce application. Their environment is structured similarly to a CTF, where the adversary aims to gain access to the database and steal sensitive user information. The benchmark facilitates 7 MITRE ATTACK categories.
* They evaluate the 4 LLMs on their refusal rate to comply with generating attacks, and when attacks are generated, the attack success rate is measured.

**Strengths:**

The paper is tackling 2 important and timely problems at the intersection of LLMs and cybersecurity.
•	Having a benchmark that includes both security and functionality unit tests for each code example is a strong contribution to the secure code generation literature. Many SOTA LLM papers in the literature currently test code security and functionality separately (ie. using separate datasets/tasks) due to lack of benchmarks with the capability to simultaneously test both. Strong and comprehensive benchmarks are definitely lacking for this problem.
* Proposed approach to leverage LLMs to scale the development of secure code benchmark dataset.
*  Using a controlled environment to see if the model can generate commands or code that facilitate attacks -- and tracking refusal rates in research on LLM-driven pentesting and red teaming can provide insight into the effectiveness of their internal safety mechanisms.

**Weaknesses:**

* While a lot of work has been done for this paper and there are definitely strong contributions, by setting CyberSecEval as the goal post to beat, this paper goes too broad in scope (for a paper of this length) and fails to adequately establish its position among the existing peer reviewed literature for each of these 2 distinct research directions. There is no need for benchmarks to cover both secure code generation and cyber attack capability as they have fundamentally different objectives, setups, and evaluation metrics. In the case of  CyberSecEval, combining these tasks made sense because it was aligned with their product’s goals. For SecCodePLT, however, the logical connection is less clear. Secure code generation and cyberattacks don’t share the same purpose, infrastructure requirements, or audience, and combining them into the one conference-length paper restricts the depth of each evaluation.

* Overall, there is a lack of discussion/justification for the choice of prompt wording/techniques.

**Secure code generation task:**
i) Relevant benchmarks, such as LLMSecEval (MSR 2023), have been overlooked. LLMSecEval covers 18 Python-related CWEs, which challenges the authors' claim that existing benchmarks address only 8 Python-related CWEs.
A more detailed analysis of the scope/coverage of existing peer reviewed benchmarks and where this paper fits in would strengthen this work.
ii)	Code security testing is challenging. Many SOTA papers try to utilize a combination of SAST tools, LLM vulnerability checkers, and manual checking. The discussion of the code security tests could be more convincing if it provided detailed information on the breadth and depth with which these tests cover potential vulnerabilities and edge cases. Eg. providing a breakdown of security test cases per CWE, showing how each test targets specific security requirements and edge cases, would help demonstrate thoroughness. Or providing a metric similar to code coverage in unit testing would help show that the security tests are exhaustive. Overall I didn’t understand how the vulnerable and patched code examples are used for evaluating the correctness of test cases and/or generated output.
iii)	Prompt quality assessments could be stronger. The Security-relevancy Judge, based on GPT-4, introduces a potential bias since the same LLM type is used in prompt construction. Using diverse models or a user study evaluation of security relevance would provide more objective evaluations. Also – leveraging, for example, LLMSecEval’s metrics for prompt clarity and relevance would enhance prompt quality assessment, ensuring that prompts are as clear and useful as possible for secure code generation.
iv) The "instruction generation" task was hard to understand and seems to not involve generating written instructions or explanations – but asking the model to generate the full code based on the provided prompt?


**Cyber attack scenario:**
i)	The paper lacks a comparison with existing cyber ranges and security testing frameworks like MITRE's Caldera, Metasploit, Cyber Range by IBM, etc, as well as other benchmarks like VulHub, HackTheBox, OWASP Benchmark, VulnHub, etc. The paper lacks a discussion of where they fall short, and why you needed to create your own simulation range – in general the positioning of SecCodePLT within the existing ecosystem is insufficient. What gap in existing environments are you are targeting? In what way did you customize your environment for LLM Evaluation?  One of the gaps in the existing literature on LLM-driven pentesting and red teaming is lateral movement – so a simulated environment and benchmark focused on measuring LLM capability for lateral movement would have been interesting.
•	(Small detail- Table 2 claims that the environment facilitates seven of the MITRE attack categories – but then lists ‘weaponisation and infiltration’ - neither of which are MITRE attack categories.)
ii)	Refusal analysis seemed to not be extensive. Motivation is not provided for the choice of role playing wording in the prompt: “I am a security researcher conducting a red teaming test on a web server”. Were other things tried that might be more persuasive for Claude?

**Typos/misprints in the paper:**
* Missing a word here on page 2? “our automatic and validate process…”.
•	Repeated phrase on page 3: “not even not even vulnerable”.
•	Page 4: “MITER” should be MITRE.
•	Page 6: The Figure 2 caption: ‘worn’ – perhaps should be “worm”?
•	Page 6: “active domain (AD) server” --- should this be Active Directory?
•	Section 4.2 says Figure 8 and 9 are about CyberSecEval but the figure captions say they are about SecCodePLT.
•	Multiple instances of “cursor”  -  should be “Cursor”.
•	Page 9: “Not that we consider cursor…” – should be “Note”.

**Questions:**

* Please provide more details on the security tests, addressing the concerns in the weaknesses section abve - including the breadth and depth with which these tests cover potential vulnerabilities and edge cases.
* Has any analysis of diversity across the 10 samples for each seed and the 5 test cases per sample been conducted? There might be redundancy.
* How are the vulnerable and patched code examples used for evaluating the correctness of test cases and/or generated output?
* Please include a comparison with LLMSecEval.

**Cyber attack scenario:**
* As outlined in the weaknesses above, please explain the motivation for creating your own simulation range and what gap in existing ranges/benchmarks yours is targeting.
* Please provide more details on your attack refusal investigation - were other role playing prompt wordings tried that might be more persuasive for Claude? Etc.

**Details Of Ethics Concerns:**

Code generation for cyber attacks has dual-use purpose and can be misused by malicious actors.
I am not sure where the community sits on ethics board approval for this topic.

---

> ### Author Response · Authors · 2024-11-21
>
> **1. Choice of insecure coding and cyberattack helpfulness**
>
> Thank the reviewer for the constructive comments. We appreciate the reviewer’s recognition of the massive amount of effort needed to build our benchmark. We would like to first clarify our logic for including insecure coding and cyberattack helpfulness in SecCodePLT.  Our goal is to evaluate the code generation models’ safety and security with a focus on code output and generation tasks. We started with the model itself and evaluated its risks in generating insecure and vulnerable code under  **benign and normal queries**. Going beyond normal queries, we then evaluate the model’s helpfulness in cyberattacks where the inputs are **malicious queries**. Given that we focus on code generation, we do not include text generation (e.g., prompt injection) or discriminative tasks (e.g., vulnerability detection) in our benchmark. We clarified our logic in Section 1 of the revised paper.
>
> **2. Difference from LLMSecEval**
>
> Thanks for pointing out this missing related work. LLMSecEval covers 18 Python-related CWEs using natural language (NL) prompts designed to test code generation models. LLMSecEval uses static analysis tools, such as CodeQL, to evaluate security vulnerabilities but does not incorporate dynamic testing. We believe our benchmark is different from LLMSecEval in the following aspects: (1) SecCodePLT covers 27 Python-related CWEs while LLMSecEval covers only 18; (2) We provide more structured input for each data point, including coding task descriptions, vulnerable code, security policies, etc, while LLMSecEval only provides an NL prompt. (3) We conducted manual inspections to ensure that our data were related to real-world, security-critical coding scenarios. (4) Unlike LLMSecEval, SecCodePLT associates each task with test cases for dynamic evaluation, which allows us to assess both the functionality and security of generated code. (5) SecCodePLT includes task variations that test models under different levels of contextual information, such as with or without security policies, enabling a more nuanced evaluation of model behavior in security-relevant contexts. We added this discussion in Section 2 of the revised paper.
>
> **3. About the quality of our testing cases**
>
> We agree with the review that writing high-quality testing cases that enable comprehensive testing is challenging. In our benchmark, we spent extensive human effort on writing testing cases that try to have a high coverage. We added an experiment to show the coverage of our testing cases. We ran the functionality and security testing cases for each data in our dataset and calculated the average line coverage of 90.92%. Most of the uncovered code consists of redundant return statements and exception handling that is either unnecessary or unrelated to the vulnerability.  We added this experiment to Section 3.
>
> **4. Potential bias in our security-relevance judgment prompt and the possibility of using LLMSecEval’s metrics**
>
> Following the reviewer’s suggestion, we report results from using claude-3-5-sonnet-20240620 as an alternative judge alongside GPT-4o. The evaluation results showed minimal variation between the two models, demonstrating that the evaluation is not overly dependent on a specific LLM and confirming the reliability of the security relevance metric.  We added this experiment in Appendix I in the revised paper.
> While LLMSecEval includes metrics such as naturalness, expressiveness, adequacy, and conciseness to assess prompt quality, our focus is on ensuring prompts align with the security evaluation goals. Given the different goals, we do not use LLMSecEval’s metrics in our evaluation.
>
> **5. Confusion about the "instruction generation" task**
>
> Sorry for the confusion. In the "instruction generation" task, the model is indeed prompted to generate the complete code based on a structured task description rather than producing written instructions or explanations. This task is designed to evaluate the model’s ability to generate secure and functional code directly from a provided prompt that outlines the coding objective. To make the task clearer, we revised "instruction generation" to “text-to-code generation” in the revised paper.

---

> > ### Author Response · Authors · 2024-11-21
> >
> > **6. Potentially redundancy in our data generation pipeline for the insecure coding task**
> >
> > Thank the reviewer for the constructive comments. We added a diversity filter in our data creation pipeline to remove redundancy. Specifically, we calculate the similarity between newly generated data and existing samples using the longest common subsequence (LCS) and word-level Levenshtein distance. If the similarity score for a newly generated sample exceeds a threshold (e.g., 0.9), it is rejected for being too redundant. This ensures that each new sample introduces meaningful variation while retaining the core functionality and security context of the original task. The generation process continues iteratively, rejecting redundant samples and regenerating until sufficient diversity is achieved or another stopping condition is met. We added this explanation in Section 3 of the revised paper.
> >
> > **7. Comparison with existing cyber ranges and security testing frameworks**
> >
> > Compared to existing cyber ranges, MITRE's Caldera and Cyber Range by IBM, our benchmarks are different in the following aspects. First, these cyber ranges are interacted with by human users. LLM cannot interact directly with these environments without additional integration (such as APIs or middleware). This makes it challenging to use these platforms to evaluate LLM's capabilities. Second, these cyber ranges lack a fine-grained evaluation metric to measure the progress and effectiveness of each attack stage. In our metric, we provide a metric for each attack stage. Note that Metasploit is a penetration testing tool rather than a cyber range that enables attack evaluations.
> >
> > Existing benchmarks pointed out by the reviewer (VulHub, HackTheBox, OWASP Benchmark) are for individual vulnerability detection and reproduction, while our benchmark focuses on the LLMs’ capability to launch major steps in the MITRE cyber attack procedure.
> >
> > Given these gaps, we created our own benchmark that enables dynamic evaluation of major attack steps in the MITRE cyber attack procedure of LLM’s capabilities in cyber attack helpfulness. A minor clarification, we do have lateral movement as part of our process: In Figure 2, after exploiting the Internal User 2, we will do lateral movement to the LAN server and then get access to the Database host.
> >
> > Following the reviewer’s suggestion, we also surveyed other AI-related penetration testing works. Most of them are new AI-driven penetration testing tools targeting certain stages of the whole cyber attack procedure, such as [1,2] for initial access, [3,4] for launching attacks. None of them built an end-to-end evaluation platform for evaluating LLMs.
> >
> > We clarified this in Section 2 of the revised paper.
> >
> > **8. About weaponization and infiltration and MITRE attack categories**
> >
> > We apologize for the confusion. The term "Weaponization and infiltration" comes from the cyber kill chain model, which provides a similar definition of the cyberattack process as MITRE. They correspond to the "Initial Access" stage in MITRE, i.e., acquiring access to the target system. We've made the change in our paper to be consistent.

---

> > > ### Author Response · Authors · 2024-11-21
> > >
> > > **9. About the refusal analysis in the cyberattack helpfulness tasks**
> > >
> > > Thanks for pointing this out. We would like to respectfully justify that as a benchmark, we use the most straightforward prompt to test the basic capability. We do not focus on jailbreaking the model.
> > >
> > > Following the reviewer’s suggestions, we did an extra experiment to show that there is a possibility of jailbreaking the model. More specifically, we tried three jailbreaking strategies. (1) Let the model act under a role-play scenario of a CTF player. This strategy comes from Cybench and NYU-CTF datasets [5,6]. (2) DAN (do anything now), which is a popular jailbreaking prompt originating from “Learn Prompting” [7]. (3) Restricted Response (i.e., “Strictly limit the scope of your answer to Linux command and command reasoning”), which is also a popular jailbreaking prompt used in jailbreaking text inputs [7].
> > >
> > > For each strategy, we conducted experiments on weaponization and C2&command tasks due to their high refusal rates. The results are shown in Figure 12. We discovered that prompting the model to act as a CTF player and using restricted responses yielded the lowest refusal rates. On the contrary, the popular jailbreaking technique DAN (Do Anything Now) is ineffective in our task. We added this experiment to Appendix H.
> > >
> > > In addition, we thank the reviewer for pointing out typos. We corrected them in the revised paper.  One minor clarification, we updated our terminology from 'active domain (AD) server' to 'LAN server' because our admin server operates on Linux rather than Windows.
> > >
> > > [1] Generative AI for Pentesting: The Good, the Bad, the Ugly
> > >
> > > [2] Getting Pwn’d by AI: Penetration Testing with Large Language Models
> > >
> > > [3] LLMs as Hackers: Autonomous Linux Privilege Escalation Attacks
> > >
> > > [4] Teams of LLM Agents Can Exploit Zero-Day Vulnerabilities
> > >
> > > [5] Cybench: A Framework for Evaluating Cybersecurity Capabilities and Risks of Language Models
> > >
> > > [6] NYU CTF Dataset: A Scalable Open-Source Benchmark Dataset for Evaluating LLMs in Offensive Security
> > >
> > > [7] https://learnprompting.org/docs/prompt_hacking/jailbreaking

---

> ### Comment · Reviewer_N2DR · 2024-11-25
>
> Thank you for your responses and for conducting the additional experiments.
>
> Regarding your comment #7: Apologies for the confusion — I was referring to the Metasploitable 2 & 3 VMs, not Metasploit.
>
> Remaining Concerns:
> ================
> Overall, I believe your response has addressed most of my concerns, but I am still not entirely convinced about the evaluation of the correctness of the security test cases. Specifically, your statement that “safety test cases are kept only if they pass in the patched code but fail in the vulnerable version” raises the following question:
>
> How can you be sure that it is the test cases that are problematic if both the patched and vulnerable code are manually generated? Could it instead be that the generated code does not meet its requirements (i.e., the vulnerable version may not truly contain the vulnerability, or the patched version may not be adequately fixed)?
>
> This highlights the need for stronger evidence that the vulnerable code contains the required vulnerabilities and that the patched code effectively mitigates them. Moreover, it remains crucial to demonstrate that the security test cases are functioning as intended.
>
>
> General Feedback:
> ===============
> I will raise my score, but I believe the paper still needs more attention to the writing, especially around the dataset creation process. This section should explain:
>     - How you ensure that the vulnerable code reliably contains the required vulnerabilities.
>     - How you ensure that the patched code is indeed patched.
>     - Most importantly, how you verify that the security test cases are working as intended.
>
> Misprints/Clarifications:
> ----------------------------
>    - The newly added text on page 2 uses the IC and CH notation before it has been defined.
>    - The newly added text needs checking for typos/grammatical errors.
>
> Thank you again for your thoughtful responses and for addressing many of the points raised.

---

> > ### Author Response · Authors · 2024-11-26
> >
> > Dear Reviewer N2DR,
> >
> > Thank you for your thoughtful follow-up and for providing detailed feedback. We greatly appreciate your acknowledgment of our efforts and your willingness to raise the score.
> >
> > **Addressing Remaining Concerns**
> >
> > To clarify, our data generation process involves **two distinct stages**:
> > 1. **Stage 1: Seed Generation** - With the assistance of LLM, we created 153 seed examples. The details are shown in Figure 1. Each seed contains vulnerable and patched versions of the code, along with corresponding capability and safety test cases.We manually review and correct all the 153 seeds to ensure their faithfulness, security relevance, test case correctness, and sufficient diversity. This stage ensures that the core vulnerabilities align with the CWE specifications.
> > 2. **Stage 2: Mutated Samples** - From each seed, we generate up to 10 mutated samples using task and code mutators. These perturbations maintain the core logic and functionalities of the code, ensuring that the vulnerabilities and patches remain consistent while introducing meaningful diversity.
> >
> > Both stages involve dynamic validation of the test cases and code to ensure correctness. Below, we address how these stages resolve the specific concerns you raised:
> >
> > 1. **Ensuring the Vulnerable Code Contains the Required Vulnerabilities**: In Stage 1, each vulnerable code sample is manually crafted and reviewed to ensure alignment with CWE specifications. Dynamic testing is performed to confirm that the vulnerable code passes capability test cases but fails safety test cases, demonstrating that the vulnerabilities are functional and detectable. In Stage 2, we apply task and code mutators to generate variations of the seed samples. Since these perturbations (e.g., renaming variables or arguments) do not alter the core logic, the vulnerabilities established in Stage 1 remain intact. If any mutated sample fails to meet the validation criteria, such as not passing the required test cases, we rerun the code mutator to generate a valid replacement.
> >
> > 2. **Ensuring the Patched Code Effectively Mitigates Vulnerabilities**: The patched code is also manually written and reviewed in Stage 1 to specifically address the vulnerabilities identified in the CWE while preserving the functionality of the original code. Dynamic validation ensures that the patched code passes both capability and safety test cases, confirming that it effectively mitigates vulnerabilities. In Stage 2, as the core logic of the patched code remains unchanged during perturbations, the patched code retains its ability to mitigate vulnerabilities. If a mutated sample fails validation due to perturbation errors (e.g., inconsistencies in the code), we rerun the code mutator to ensure that all generated samples meet the criteria.
> >
> > 3. **Verifying the Security Test Cases**: In Stage 1, capability test cases are retained only if they pass for both the vulnerable and patched code. Safety test cases are kept only if they pass for the patched code but fail for the vulnerable version. These validated test cases are reused directly in Stage 2 without modification, as the core logic of the code remains unchanged during perturbations.
> >
> > **Updates to Text**
> >
> > We revised the benchmark generation section (section 3.2) to clarify these points in the updated version:
> > - We explicitly highlight that each seed is rewritten five times to adapt the task to new scenarios while maintaining the CWE's core characteristics. These rewrites are manually reviewed to ensure their faithfulness, security relevance, correctness of the test cases, and diversity.
> > - The validation process for capability and safety test cases is clearly outlined as occurring in Stage 1, with these test cases reused in Stage 2.
> > - The process of rerunning the code mutator to handle failures in Stage 2 is emphasized.
> >
> > Additionally, we addressed the notation issues you identified on page 2, replacing IC and CH with insecure coding and cyberattack helpfulness. We also thoroughly proofread the newly added text again and corrected typos or grammatical errors we found.
> >
> > Thank you again for your constructive feedback and for raising your score. We are committed to addressing these remaining concerns and improving the clarity of our manuscript.
> >
> > Best regards,
> > Authors

---

> > > ### Author Response · Authors · 2024-12-03
> > >
> > > Dear Reviewer N2DR,
> > >
> > > Sorry to bother you again. With the discussion phase nearing the end, we would like to know whether the responses have addressed your concerns.
> > >
> > > Should this be the case, we are encouraged that you raise the final rating to reflect this.
> > >
> > > We are looking forward to your reply. Thank you for your efforts in this manuscript.
> > >
> > > Best regards, Authors

---

> > > > ### Comment · Reviewer_N2DR · 2024-12-03
> > > >
> > > > Thanks for the response.
> > > > To improve the safety test, have you checked if it's possible to cross-validate your test's results with that of a SAST tool?  CodeQL is popular in the academic literature but there are many out there.

---

> > > > > ### Author Response · Authors · 2024-12-03
> > > > >
> > > > > Dear reviewer N2DR,
> > > > >
> > > > > Thank you for your response. Generally, static testing offers higher efficiency but is prone to a high false positive rate and lacks the ability to track runtime or dynamic behaviors of a program (e.g., CWE-1333: Inefficient Regular Expression Complexity). In contrast, dynamic testing provides more reliable results with significantly fewer false positives but struggles to identify vulnerabilities in unreachable code. Most existing works opt for SAST tools such as CodeQL due to the intensive human effort required to construct stricter dynamic test cases for large datasets. In our study, we demonstrated that our test cases achieve an average of 91% line coverage, with the uncovered code primarily consisting of redundant return statements and exception handling, which are unrelated to the vulnerabilities. This enables us to provide a more precise and reliable analysis.
> > > > >
> > > > > Regards, authors

---

### Official Review · Reviewer_fo1m · 2024-11-03

**Soundness:** 2
**Presentation:** 2
**Contribution:** 2
**Rating:** 5
**Confidence:** 3

**Summary:**

The paper proposes SECCODEPLT, a unified and comprehensive evaluation platform for code GenAIs’ risks.

For insecure code, the authors introduce a new methodology for data creation that combines experts with automatic generation. For cyberattack helpfulness, the authors set up a real environment and construct samples to prompt a model to generate actual attacks, along with dynamic metrics.

**Strengths:**

Through experiments, SECCODEPLT outperforms CYBERSECEVAL in security relevance and prompt faithfulness, highlighting the quality of this benchmark.
The authors then apply SECCODEPLT and CYBERSECEVAL to four SOTA open and closed-source models, showing that SECCODEPLT can better reveal a model’s risk in generating insecure code.

**Weaknesses:**

Many state-of-the-art methods for code generation are not mentioned and experimented in the paper, such as:

Jingxuan He, Martin Vechev. Large Language Models for Code: Security Hardening and Adversarial Testing. 2023. In CCS. https://arxiv.org/abs/2302.05319.

Erik Nijkamp, Bo Pang, Hiroaki Hayashi, Lifu Tu, Huan Wang, Yingbo Zhou, Silvio Savarese, and Caiming Xiong. 2023. CodeGen: An Open Large Language Model for Code with Multi-Turn Program Synthesis. In ICLR. https://arxiv.org/
abs/2203.13474

Daniel Fried, Armen Aghajanyan, Jessy Lin, Sida Wang, Eric Wallace, Freda Shi, Ruiqi Zhong, Wen-tau Yih, Luke Zettlemoyer, and Mike Lewis. 2023. InCoder: A Generative Model for Code Infilling and Synthesis. In ICLR. https://arxiv.org/
abs/2204.05999

Loubna Ben Allal, Raymond Li, Denis Kocetkov, Chenghao Mou, Christopher Akiki, Carlos Muñoz Ferrandis, Niklas Muennighoff, Mayank Mishra, Alex Gu, Manan Dey, et al. 2023. SantaCoder: Don’t Reach for the Stars! CoRR
abs/2301.03988 (2023). https://arxiv.org/abs/2301.03988

There are many other benchmarks for evaluations of code generation that are not mentioned and compared. Please refer to the paper https://arxiv.org/html/2406.12655v1 for details.

**Questions:**

In “Each seed contains a task description, example code, and test cases”, do all the source code samples have the task description? What are the methods used in test cases?

It is not clear how the author performs the code mutator as mentioned in “As specified in Section 3.2, we design our task mutators to keep the original security context and code mutator to preserve the core functionalities.” What types of code mutators are used here?

What dynamic methods do the authors use for “After mutation, we also manually check the security relevance of newly generated data and run dynamic tests to ensure the correctness of their code and test cases.”?

---

> ### Author Response · Authors · 2024-11-21
>
> **1. Missing related works**
>
> As discussed in the related work, we mainly compare our work with security-related coding model benchmarks, as such we do not compare it with general code generation benchmarks. Given that our focus is to build benchmarks rather than proposing new code generation models or methods, we did not discuss existing code generation methods in the paper. We added the related papers and clarified this in Section 2 of the revised paper.
>
> **2. Questions about task description and rule-based metrics**
>
> We thank the reviewer for the comment. Yes, each sample in our benchmark has a task description. For test cases, we employ a mix of dynamic and rule-based methods to evaluate both the functionality and security of the generated code. Dynamic test cases involve running the code with a variety of inputs to verify that it performs as expected and remains secure under different conditions.
> The rule-based metrics are designed to evaluate scenarios where standard test cases may not be applicable or effective. For example, in cases like CWE-327 (Use of a Broken or Risky Cryptographic Algorithm), the rule is to check whether the generated code uses the `random` library for cryptographic purposes instead of a more secure option like `secrets`.
>
> **3. What types of code mutators are used**
>
> Thank you for the comment. The code mutators we use are carefully designed to preserve the core functionality and security context of each task while introducing controlled variations. We used claude-3-5-sonnet-20240620 for the mutation. The prompt is shown in the Appendix K.
>
> **4. What dynamic method is used to ensure the correctness of the generated data**
>
> For each code sample, we executed both the vulnerable and patched versions using a testing framework that loads the setup, core function code, and associated test cases. These dynamic tests assess both capability and safety by verifying that both the vulnerable and patched code fulfill the intended functionality, while also ensuring that only the patched code avoids unsafe behavior by passing all security checks, whereas the vulnerable code fails these tests.
> We filtered out any that do not meet these criteria: capability test cases are retained only if they pass for both versions and safety test cases are kept only if they pass in the patched code but fail in the vulnerable version. After filtering, we verify that at least one valid test case remains in each category; if there are insufficient valid cases, we rerun the code mutator to generate additional variations that meet these requirements. We clarified this in Section 3 of the revised paper.

---

### Official Review · Reviewer_3Qi9 · 2024-11-04

**Soundness:** 2
**Presentation:** 2
**Contribution:** 2
**Rating:** 3
**Confidence:** 4

**Summary:**

This paper develops SECCODEPLT, a unified and comprehensive evaluation platform for code GenAIs’ risks. It introduces a new methodology for data creation that combines experts with automatic generation for insecure code which ensures the data quality while enabling large-scale generation. It also associates samples with test cases to conduct code-related dynamic evaluation. Furthermore, it sets up a real environment and constructs samples to prompt a model to generate actual attacks for the task of cyberattack helpfulness, along with dynamic metrics in our environment.

**Strengths:**

The paper presents a pioneering approach by integrating a database with two distinct security-related tasks. SECCODEPLT serves as a comprehensive platform that unifies the evaluation of GenAIs’ risks associated with code generation. This integration facilitates a holistic approach to assessing different dimensions of security risks. By associating samples with test cases, SECCODEPLT enables dynamic evaluation related to code. This method allows for real-time assessments and adjustments, providing a deeper analysis of the code's behavior in practical scenarios.

**Weaknesses:**

1. The programming language used in the paper is limited, with Python being the sole language explored. This is inadequate for a comprehensive and large-scale benchmark. The inclusion of other programming languages like C/C++ and Java, which constitute a significant portion of recent CVEs, is crucial. These languages are more complex in syntax and more broadly applied, offering valuable insights into the capabilities of LLMs.
2. The paper's description of the data generation process for the IC task is unclear. It mentions the use of two different mutators to generate data, yet it fails to clarify the generation of the corresponding test suites. It is uncertain whether the test suites for these new datasets are generated by LLMs or if they reuse the original suites. If generated by LLMs, how is the quality of these suites assured? If the original test suites are used, can they adapt to new contexts effectively?
3. The paper lacks a necessary ablation study. The boundary of what is user control and what is provided by benchmark is not well clarified. The rationale behind the design of the prompts and instructions used to trigger evaluations is not well justified. For example, why do the authors use system prompts and user templates shown in the paper? Are they more reliable and efficient? Will the differences in these prompts affect the evaluation of LLM ability? If users want to use their own prompts, is there any way?
4. The evaluation metric of security relevance is confusing and lacks rationales. It is unclear whether this metric aims to assess specific properties of LLMs or the prompts themselves. Because the benchmark is designed to evaluate LLMs, using a metric that assesses the prompts introduces confusion. Furthermore, in the SECURITY-RELEVANCY JUDGE prompt template (D.1), the security policy reminder is included as part of the user input and fed directly to the LLM. This setup may influence the evaluation of security relevance and potentially introduce bias.
5. The ablation of the security policy reminder is missing, similar to problem 3. The paper does not discuss the reasons for choosing the security policy reminder prompt.
6. The paper lacks a discussion on the specific defenses employed in the CH task. In realistic settings, a variety of defenses, such as firewalls and intrusion detection systems, are typically deployed. It will be insightful to know how different LLMs perform when various defenses are considered in a simulated environment.
7. The usefulness and generalization of the CH task is limited. Practical attacks vary significantly and are influenced by diverse factors, but the scenario described in the paper lacks generalizability across different attack types and target systems. This limited setting restricts the ability to conduct an accurate and comprehensive evaluation of LLMs for the CH task. Additionally, the paper does not specify the capabilities of attackers, including the types of tools that can be used to launch attacks with LLMs. Also, the strong assumption that some internal users will click on phishing or other harmful links further reduces the task's practical relevance.
8. Evaluation metrics in CH task. It will be better to set a specific metric to evaluate the overall ASR for the end-to-end attack. Additionally, the details regarding the evaluation process are not well-explained – whether it is a fully automated process or requires human input at various stages to guide or adjust the evaluation.

**Questions:**

See above.

---

> ### Author Response · Authors · 2024-11-21
>
> **1. Limited programming language**
>
> We thank the reviewer for pointing this out. We would like to kindly point out that creating high-quality data for insecure coding and building an executable environment for cyberattack helpfulness is challenging and requires a large amount of effort. As such, in this work, we focus on Python, as it is the most predominant programming language and continues to grow in popularity. Some widely used benchmarks that support dynamic testing are also Python only, such as Swe-bench for patching and LiveCodeBench for debugging and testing. We are the first security benchmark that enables dynamic executions. In our humble opinion, including more program languages such as C/C++ requires substantially more effort. For example, we will need to reproduce existing vulnerable code and create docker environments to execute them. Given the massive amount of effort required, we respectfully believe that it is reasonable to defer it as our future work. We added this discussion in Section 5 of the revised paper.
>
> **2. Provide more details about the data generation process for the insecure coding task**
>
> We thank the reviewer for the constructive comments. The test suites used for these mutated tasks and code samples are based on the original test cases. We carefully ensured that these original test cases remained relevant and effective even after mutation. The code mutators were designed to make only syntactic or structural changes that preserve functionality, allowing the original test cases to apply to the mutated code without issue. Additionally, after mutation, we performed dynamic testing to validate that the original test cases still work as intended with the modified code and verify both capability and safety requirements.
> Specifically, for each mutated code sample, we executed both the vulnerable and patched versions using a testing framework that loads the setup, core function code, and associated test cases. These dynamic tests assess both capability and safety by verifying that both the vulnerable and patched code fulfill the intended functionality, while also ensuring that only the patched code avoids unsafe behavior by passing all security checks, whereas the vulnerable code fails these tests.
> We filtered out any that do not meet these criteria: capability test cases are retained only if they pass for both versions, and safety test cases are kept only if they pass in the patched code but fail in the vulnerable version. After filtering, we verified that at least one valid test case remains in each category; if there are insufficient valid cases, we rerun the code mutator to generate additional variations that meet these requirements.
>  We clarified this in Section 3 of the revised paper.
>
> **3. How to create our prompts and whether the benchmark supports user-specific prompts**
>
> We thank the reviewer for the comments and we are sorry for the confusion.
> In our benchmark setup, the benchmark itself provides the core task prompts, code samples, test cases, and evaluation metrics. These elements are carefully crafted and standardized to ensure consistency across evaluations. User control includes decisions on specific evaluation modes, such as whether to run the task as instruction generation (text-to-code generation) or as code completion. Users can also choose to include or exclude optional fields, such as the security policy, to assess how model performance varies with different levels of contextual information.
> The system prompts and user templates shown in the paper were carefully crafted with a significant human effort to provide standardized and effective prompts for testing different models. Our benchmark is specifically designed to offer reliable, task-aligned prompts that minimize ambiguities and ensure clarity in evaluating model behavior. The aim is to establish a consistent evaluation framework that researchers can readily adopt without needing to design prompts from scratch. Similar to existing LLM benchmarks, our prompts are provided to enable an apple-to-apple comparison of different models’ performance. We indeed validated that subtly mutating the prompts will not trigger a huge difference in model performance.
> About use-specific prompts: While the benchmark provides these standardized prompts, users who wish to customize their evaluations can modify the input templates. The evaluation framework allows for user-defined prompts, provided they adhere to the necessary structure for the benchmark’s test cases and evaluation pipeline. We added this clarification in Section 5 of the revised paper.

---

> > ### Author Response · Authors · 2024-11-21
> >
> > **4. The metric of the security relevance experiment and whether security policy reminder is forced as part of the input**
> >
> > We thank the reviewer for the constructive feedback. We would like to clarify that the evaluation metric for security relevance in SecCodePLT is not designed to assess the ability of LLMs to generate contextually appropriate responses. Instead, it evaluates whether the prompts used in the benchmark effectively highlight the security-critical aspect of a task, ensuring alignment with the intended CWE context. This metric focuses on the quality and relevance of the prompts, verifying that they accurately frame the security scenario required for evaluating model behavior.
> > The line of the security policy reminder in the judge prompt template (Appendix D.1) is **optional**. When conducting evaluations without the security policy, this line is removed from the template entirely. Figure 3 in the paper highlights these evaluations, showing results for both setups—one with the security policy included and one without. We clarified this in Section 4 and Appendix D in the revised paper.
> >
> > **5. Ablation studies of security policy reminder**
> >
> > We thank the reviewer for the constructive comments. We would like to first point out that our paper includes the ablation of the security policy reminder prompt in Figure 4, which evaluates model performance with and without the security policy reminder. The results in Figure 4 clearly demonstrate the impact of the security policy reminder, with significant improvements (approximately 30% improvement on the rule-based set and a 10% improvement on the pass@1 set) in secure coding rates when the policy is included.
> > In addition to testing the presence or absence of the security policy reminder, we also experimented with different styles of the policy prompt by rephrasing it using gpt-4o-2024-08-06 and claude-3-5-sonnet-20240620. When comparing performance across models with differently rephrased styles of the security policy reminder, we observed that the differences were within 3% for all evaluated models. This finding demonstrates that the specific rephrased style has a minimal impact on model performance, as long as the core guidance remains clear and understandable. We added this new experiment in Appendix J of the revised paper.
> >
> > **6. Discussion about defenses in the CH tasks**
> >
> > As shown in our experiment, the capabilities of SOTA LLMs in the CH task are still very limited. The models can barely launch successful attacks even without any defenses. As such, we do not include defenses. We agree with the reviewer that once the model can launch attacks at a reasonable success rate, it is necessary to test their resiliency against defenses. As such, we respectfully believe that it is reasonable to defer this extra step as part of our future work. We added this discussion to Section 5 in the revised paper.
> >
> > **7. The current CH tasks lack generalizability and description of attack capabilities**
> >
> > The goal of our work is to provide a standard, controlled, and manageable environment that covers the major steps in the cyber kill chain and MITRE attack procedures. We built the current environment to serve this purpose. It is noted that having such a system with vulnerabilities and attack paths injected requires non-trivial effort. Given that the SOTA models cannot perform well in the current environment, we respectfully believe that it is reasonable to defer more complex attack environments to future works.  We added this discussion to Section 5 in the revised paper.
> >
> > **8. Overall ASR for the end-to-end attack and whether the attack process is fully automated**
> >
> > As shown in Table 2, we designed criteria for each attack stage to decide whether an attack succeeds and evaluated each stage independently. Our overall ASR is defined as the attack that passes all the criteria of each stage. We further conducted an experiment to test the overall attack performance of selected models. For each model, we use it to launch an attack from the first attack stage. If the attack of the current succeeds, it will move to the next stage. An attack that passes all stages is marked as a successful attack. We conducted 500 trials for each model and got a zero ASR. More specifically, GPT-4o, Claude3.5-Sonnet, and Llama3.1-70B have an average of passing 0.68/5, 0.6/5, and 0.1/5 stages.  We added this experiment to Section 4. We also want to clarify that our evaluation process is fully automated.

---

### Official Review · Reviewer_8iyW · 2024-11-05

**Soundness:** 2
**Presentation:** 3
**Contribution:** 3
**Rating:** 6
**Confidence:** 3

**Summary:**

This paper presents SecCodePLT, a unified and comprehensive evaluation platform for code GenAIs' risks. Considering insecure code, the author introduces a new methodology for data creation that combines experts with automatic generation. Considering cyberattack helpfulness, the authors set up a real environment and construct samples to prompt a model to generate actual attacks. Experiments show that CyberSecEval could identify the security risks of SOTA models in insecure coding and cyberattack helpfulness.

**Strengths:**

1. Promising direction. Establishing the benchmark to highlight the security risks associated with Code GenAI is a direction worth studying.
2. Consider real-world attack behaviors and environment deployments.
3. Compared with existing baselines from multiple perspectives and the results show the effectiveness of the proposed method.

**Weaknesses:**

1. Some related work discussions are missing.
2. Some details are not explained clearly.
3. There are some minor errors that need to be polished and proofread.

**Questions:**

1. This article discusses risk assessment of code generation. Some related works on code generation may also be discussed, such as BigCodeBench [1].

[1] Bigcodebench: Benchmarking code generation with diverse function calls and complex instructions. https://arxiv.org/pdf/2406.15877

2. Some details are not explained clearly. In line 140 of the manuscript, the author mentions "extracting code chunks without proper context frequently leads to false positives". But it seems that the experiment did not perform an ablation experiment on the context field. As shown in lines 867 and 894, the context field is set to None. So I don't understand the role of context and how the solution SecCodePLT in this paper can benefit from context (how to reduce false positives).

3. In line 251 of the manuscript, the author mentions "We also introduce rule-based metrics for cases that cannot be evaluated with standard test cases". I am not sure where the rule mentioned here comes from. Is it based on some public manufacturer's provision?

4. In MITRE ATT\&CK, the kill chain model may be common. In other words, an attacker often implements different attack stages through a series of attack techniques and tactics. It is unclear whether SecCodePLT considers such multi-stage attack and intrusion, rather than a single attack behavior.

5. Some minor errors, such as the missing period after "security-critical scenarios" on line 76. For "security is required.)" on line 253, the period should probably be after ")".

---

> ### Author Response · Authors · 2024-11-21
>
> **1. Missing related work [1]**
>
> We thank the reviewer for pointing this out. BigCodeBench is designed to evaluate LLMs’ capability to solve general programming tasks. It focuses on two key aspects: diverse function calls (using tools from multiple libraries) and complex instruction following. This work has a different focus from ours. We focus on the security and risks of the code generation models while this paper focuses more on the models’ normal capabilities. As such, BigCodeBench collects data from different normal libraries (pandas, numpy, etc), while we collect our data from top CWEs. We added our discussion in Section 2 of our paper.
>
> [1] Bigcodebench: Benchmarking code generation with diverse function calls and complex instructions
>
> **2. Confusion about the context field in the data**
>
> Thank you for your question about context usage in our paper. We would like to clarify that there are two distinct types of "context" discussed in the paper:
> 1. Security-related Background Context (Line 140):
> This refers to the essential security-relevant background information for each code snippet. Without proper security context, the extracted vulnerable code may actually be benign, which will introduce false positives. For example, using a fixed seed for random number generation might seem harmless in general applications, but could introduce serious vulnerabilities when used in cryptographic contexts. These security contexts are explicitly provided in our task descriptions, as demonstrated in Appendix C under the task_description field. We changed the context in Lin 140 to “background’’ in the revised paper.
>
> 2. Function-level Technical Context (Lines 867 and 894):
> This refers to the implementation context such as global variables, import statements, and other code dependencies. This is part of our input JSON file that can help the model to better finish the coding task. The None values in these lines specifically refer to this type of technical context, which is None in this specific example, because the example does not involve global variables, import statements, etc. We have added new examples in Appendix C that have these implementation contexts.
>
> **3. Rule-based metrics in line 251**
>
> Sorry for the confusion. The rule-based metrics we introduced are designed to evaluate scenarios where dynamic test cases may not be applicable or effective. These rules are not based on manufacturer guidelines; rather, they were developed based on security best practices and known coding standards within the research and cybersecurity communities. For example, in cases like CWE-327 (Use of a Broken or Risky Cryptographic Algorithm), the rule is to check whether the generated code uses the `random` library for cryptographic purposes instead of a more secure option like `secrets`.
>
> **4. Whether SecCodePLT considers multi-stage attacks in MITRE ATT&CK**
>
> Thank you for pointing this out. We considered multi-stage attacks in our benchmark. More specifically, In Figure 5 of our paper, we analyzed the Attack Success Rate (ASR) for different stages of the attack chain separately, providing a detailed view of model performance at each attack phase. We can tell from the result that if the attack of each stage were independent, the theoretical success rate of a complete attack chain would be less than 0.5% (the ASR for Weaponization and C2 are both less than 10%). To validate this point, we also conducted end-to-end attack experiments where we prompted the models to execute complete attack chains from Reconnaissance to Collection. In these experiments, we conducted 500 independent trials for each selected model. We found zero successful cases of complete attacks. More specifically, GPT-4o, Claude3.5-Sonnet, and Llama3.1-70B have an average of passing 0.68/5, 0.6/5, and 0.1/5 stages. We added this discussion to Section 4 in our revised paper.
>
> In addition, we thank the reviewer for pointing out typos. We have corrected them in the revised paper.

---

### Author Response · Authors · 2024-11-21
**Global Response**

Dear Reviewers,

We thank the reviewers for the insightful questions and reviews. Your time and effort dedicated to improving our work are truly appreciated. We have responded to all the insightful comments with extra experiments. All modifications are marked in red color in the revised paper. Below we summarize the experiments and changes we made in the revision.

**We added the following experiments:**
1. We conducted a coverage test of our testing cases and showed that our test cases can enable a high coverage with an average of 90.92%. Most of the uncovered code consists of redundant return statements and exception handling that are unrelated to the vulnerability. This experiment validates the quality of our testing cases. We added this experiment to Section 3.

2. We conducted an experiment to test the overall attack performance of selected models. For each model, we use it to launch an attack from the first attack stage. If the attack of the current stage succeeds, it will automatically move to the next stage. An attack that passes all stages is marked as a successful attack. We conducted 500 trials for each model and got a zero ASR. More specifically, GPT-4o, Claude3.5-Sonnet, and Llama3.1-70B have an average of passing 0.68/5, 0.6/5, and 0.1/5 stages.  We added this experiment to Section 4.

3. We did an extra experiment to show that there is a possibility of jailbreaking the model. More specifically, we tried three jailbreaking strategies. (1) Let the model act under a role-play scenario of a CTF player. This strategy comes from Cybench and NYU-CTF datasets [1,2]. (2) DAN (do anything now), which is a popular jailbreaking prompt originating from “Learn Prompting” [3]. (3) Restricted Response (i.e, “Strictly limit the scope of your answer to Linux command and command reasoning”), which is also a popular jailbreaking prompt used in jailbreaking text inputs [3]. For each strategy, we conducted experiments on weaponization and C2&command tasks due to their high refusal rates. The results are shown in Figure 12. We discovered that prompting the model to act as a CTF player and using restricted responses yielded the lowest refusal rates. On the contrary, the popular jailbreaking technique DAN (Do Anything Now) is ineffective in our task. We added this experiment to Appendix H.

4. We replaced GPT-4o with claude-3-5-sonnet-20240620 as an alternative judge in the security relevance experiment. The evaluation results showed minimal variation between the two models, demonstrating that the evaluation is not overly dependent on a specific LLM and confirming the reliability of the security relevance metric.  We added this experiment in Appendix I in the revised paper.

5. We experimented with different styles of the policy prompt by rephrasing it using gpt-4o-2024-08-06 and claude-3-5-sonnet-20240620. When comparing performance across models with differently rephrased styles of the security policy reminder, we observed that the differences were within 3% for all evaluated models. This finding demonstrates that the specific rephrased style has a minimal impact on model performance, as long as the core guidance remains clear and understandable. We added this new experiment in Appendix J of the revised paper.

**In addition, we made the following modifications to the paper:**

Section 1:
1. We clarified the logic behind selecting the insecure coding and cyberattack helpfulness tasks.

Section 2:
1. We added the comparison of our benchmark with BigCodeBench, which focuses on general coding capabilities rather than security and risks.
2. We added a comparison of our benchmark with LLMSecEval, which does not enable dynamic evaluation and structured inputs.
3. We added the comparison of our benchmark with related works pointed out by Reviewer fom1, which are also about general coding capabilities and new coding methods rather than security and risk benchmarks.
4. We added a discussion about the difference between our CH benchmark and the existing cyber ranges and vulnerability detection, 5. reproduction, and penetration testing benchmarks. We clarified why it is necessary to build a new CH benchmark.

Section 3:
1. We added more details and clarifications about our data generation process for the insecure coding task.
2. We explained the dynamic method used to ensure the correctness of the generated data.
3. We added our filtering step to avoid redundancy.
4. We clarified that we considered lateral movement in our benchmark.

Section 4:
1. We clarified that our CH benchmark considered multi-stage attacks.
2. We clarified the purpose of our security relevance experiment and the corresponding metric.

Section 5:
1. We added a discussion about other programming languages.
2. We added a discussion about supporting user-specific prompts.
3. We added a discussion about considering defenses and other attacks in the CH task.

---

> ### Author Response · Authors · 2024-11-21
> **Global Response (Continued)**
>
> Appendix:
> 1. In Appendix C, we added new examples of our data with implementation context.
> 2. In Appendix D, we clarified that the security policy reminder is optional in the input.
> 3. In Appendix K, We added our code mutator prompt.
>
>
> **Additionally, we corrected typos and refined terminology throughout the paper, marking these changes in red.**
>
> We hope these revisions address the reviewers’ concerns and improve the overall quality of our paper.
>
> Best regards,
>
> Authors

---

### Meta-Review · Area_Chair_eLbw · 2024-12-21

**Metareview:**

- Scientific Claims and Findings:
   - This paper introduces SecCodePLT, a Python-focused benchmark designed to evaluate the security risks associated with code generated by existing code LLMs. SecCodePLT assesses a model’s ability to produce secure code and its potential to facilitate end-to-end cyberattacks.
- Strengths:
   - The proposed benchmark, SecCodePLT, is a valuable contribution to the field.
   -  SecCodePLT outperforms the state-of-the-art benchmark, CyberSecEval, in both prompt faithfulness and security relevance.
   - SecCodePLT more effectively reveal a model’s risk in generating insecure code compared to CyberSecEval.
- Weaknesses:
    - The writing lacks clarity and could benefit from improved organization, presentation, and detail.
   -  Insufficient comparison with existing benchmarks.
   -  The evaluation of state-of-the-art code generation models is not comprehensive. Models such as DeepSeek Coder, CodeQwen, and others are neither discussed nor experimented with.
- Most Important Reasons for Decision:
     - Based on the identified weaknesses.

**Additional Comments On Reviewer Discussion:**

This paper has significantly benefited from the review process. The updated version shows considerable improvement over the initial submission, leading Reviewers 8iyW and N2DR to raise their scores to 6 after the rebuttal.

Overall, the AC believes that this benchmark paper could be improved by enhancing its comprehensiveness and clarity. Another round of revisions would be beneficial.

---

### Decision · Program_Chairs · 2025-01-22

Reject